# The Utility of Conventional CT, CT Perfusion and Quantitative Diffusion-Weighted Imaging in Predicting the Risk Level of Gastrointestinal Stromal Tumors of the Stomach: A Prospective Comparison of Classical CT Features, CT Perfusion Values, Apparent Diffusion Coefficient and Intravoxel Incoherent Motion-Derived Parameters

**DOI:** 10.3390/diagnostics12112841

**Published:** 2022-11-17

**Authors:** Milica Mitrovic-Jovanovic, Aleksandra Djuric-Stefanovic, Keramatollah Ebrahimi, Marko Dakovic, Jelena Kovac, Dimitrije Šarac, Dusan Saponjski, Aleksandra Jankovic, Ognjan Skrobic, Predrag Sabljak, Marjan Micev

**Affiliations:** 1Department of Digestive Radiology, Center for Radiology and Magnetic Resonance Imaging, University Clinical Center of Serbia, Pasterova 2, 11000 Belgrade, Serbia; 2Faculty of Medicine, University of Belgrade, Dr. Subotica 8, 11000 Belgrade, Serbia; 3Department of Surgery, First University Surgical Clinic, University Clinical Center of Serbia, Koste Todorovica 6, 11000 Belgrade, Serbia; 4Faculty of Physical Chemistry, University of Belgrade, 11000 Belgrade, Serbia; 5Department of Pathology, First University Surgical Clinic, University Clinical Center of Serbia, Koste Todorovica 6, 11000 Belgrade, Serbia

**Keywords:** gastrointestinal stromal tumors (GIST), CT perfusion, incoherent motion (IVIM), apparent diffusion coefficient (ADC)

## Abstract

Background: The role of advanced functional imaging techniques in prediction of pathological risk categories of gastrointestinal stromal tumors (GIST) is still unknown. The purpose of this study was to evaluate classical CT features, CT-perfusion and magnetic-resonance-diffusion-weighted-imaging (MR-DWI)-related parameters in predicting the metastatic risk of gastric GIST. Patients and methods: Sixty-two patients with histologically proven GIST who underwent CT perfusion and MR-DWI using multiple b-values were prospectively included. Morphological CT characteristics and CT-perfusion parameters of tumor were comparatively analyzed in the high-risk (HR) and low-risk (LR) GIST groups. Apparent diffusion coefficient (ADC) and intravoxel-incoherent-motion (IVIM)-related parameters were also analyzed in 45 and 34 patients, respectively. Results: Binary logistic regression analysis revealed that greater tumor diameter (*p* < 0.001), cystic structure (*p* < 0.001), irregular margins (*p* = 0.007), irregular shape (*p* < 0.001), disrupted mucosa (*p* < 0.001) and visible EFDV (*p* < 0.001), as well as less ADC value (*p* = 0.001) and shorter time-to-peak (*p* = 0.006), were significant predictors of HR GIST. Multivariate analysis extracted irregular shape (*p* = 0.006) and enlarged feeding or draining vessels (EFDV) (*p* = 0.017) as independent predictors of HR GIST (area under curve (AUC) of predicting model 0.869). Conclusion: Although certain classical CT imaging features remain most valuable, some functional imaging parameters may add the diagnostic value in preoperative prediction of HR gastric GIST.

## 1. Introduction

Gastrointestinal stromal tumors (GIST) are the most common mesenchymal neoplasms of the gastrointestinal tract. They originate from the interstitial cells of Cajal and may occur anywhere along digestive tract [1]. GISTs are found most often in the stomach (50–60%), the small intestine (30–35%) and rarely in the esophagus, colon and rectum (5%) or at the other sites of peritoneal cavity [2].

These tumors have malignant potential, and surgical resection remains the only curative therapeutic modality of localized and primary resectable GISTs [3]. However, 40% of resectable patients develop local recurrence or metastases despite complete surgical resection [4]. Thus, adjuvant treatment with the imatinib mesylate is recommended for patients with high-risk GIST. Imatinib is also a therapy of choice in the case of disease recurrence or when a tumor is unresectable [5]. European Society for Medical Oncology (ESMO) and National Comprehensive Cancer Network (NCCN) guidelines recommend adjuvant therapy for up to 3 years for patients with high risk and intermediate risk GISTs [6]. Unfortunately, the majority of patients with the advanced GIST develop disease progression during long-term imatinib treatment [6,7]. Thus, the precise preoperative risk assessment of GIST has become very important for the appropriate strategy of treatment planning [3].

There are several risk-stratification systems that use prognostic factors for recurrence such as tumor size, mitotic rate, tumor location and stage of the disease [8,9,10,11,12]. The National Institutes of Health consensus criteria (NIHC) are based on tumor size and mitotic count [8]. In addition to NIHC, the Armed Forces Institute of Pathology Criteria (AFIPC) uses tumor localization as third prognostic factor [9]. Modified NIHC classification includes both of the above classifications’ criteria but adds a tumor rupture factor [10,11]. On the other hand, the American Joint Committee on Cancer staging system uses the TNM classification [12].

It cannot be clearly stated which classification system is the best for the selection of patients for adjuvant therapy. All these indicators are mainly post-surgical indexes, so it would be of great clinical importance to obtain a non-invasive method that could be used to evaluate the malignant potential of GISTs preoperatively.

The gold standard for diagnosing GIST is computed tomography (CT) with high sensitivity in tumor detection and staging [12]. CT perfusion provides information on tissue vascularization by evaluating the dynamics of contrast medium distribution, while magnetic resonance (MR) diffusion-weighted imaging (DWI) provides qualitative and quantitative data on tissue cellularity based on random water molecule diffusion [13,14]. 

MR-DWI enables the clear visualization of tumor but also quantitative evaluation through the apparent diffusion coefficient (ADC) measurements. Intravoxel incoherent motion (IVIM) imaging enables the separation and evaluation of the contributions of perfusion and true molecular diffusion in the form of true diffusion coefficient (D slow), pseudo-diffusion coefficient (D fast) and perfusion fraction (f) by using multiple b-values according to a bi-exponential model [14].

However, to our best knowledge, no study has comparatively investigated the role of functional imaging, such as CT perfusion, IVIM and DWI-ADC, along with the tumor’s morphological CT characteristics in the preoperative prediction of the metastatic potential of GISTs. Thus, the aim of this study was to investigate the potential role of standard and advanced CT and MR imaging techniques as classical CT features, CT-perfusion, DWI-ADC and IVIM-related parameter values in a non-invasive prediction of metastatic risk of gastric GISTs.

## 2. Materials and Methods

Sixty-two patients with histopathologically confirmed GIST of the stomach who underwent regular abdominal CT examination with the CT perfusion and diffusion MRI from February 2019 to June 2022 were included in this single-center prospective study. Inclusion criteria included the following: (1) the clinical suspicion of newly discovered gastric submucosal lesion based on the findings of previous examinations, (2) CT exam with the CT perfusion, (3) MR exam with the DWI, (4) operated and histologically and immunohistochemically confirmed GIST, (5) the time period between the CT and MR-DWI examination being no longer than 7 days, and (6) a time gap from the performed diagnostics to the surgery of up to 20 days.

Exclusion criteria were: (1) non-gastric GIST, (2) other types of gastric tumors revealed by post-operative histopathology, (3) patients whose CT or MR-DWI image quality was reduced due to artifacts so that further post-processing analysis was unfeasible, and (4) exceeding the above-specified time period between two diagnostic modalities or between performed diagnostics and the surgery.

Finally, 62 patients with histopathologically confirmed GIST of the stomach with the low-dose CT perfusion study incorporated in the conventional CT examination were included in our study. MRI examination with DWI was not performed in all patients due to the unavailability of this diagnostic modality at the given time and contraindications for the MR examination (claustrophobia, implanted pacemaker, etc.), so 45 patients underwent MRI with DWI. In all cases, the ADC was obtained by using the mono-exponential model for the combination of b = 0 and b = 800. In 34 of those patients the whole designed diagnostic protocol was completed with IVIM parameters. In the remaining eleven patients, difficulties in processing multiple b-values diffusion measurements after fitting the signal values from manually located ROIs enabled the reliable calculation of IVIM parameter values. 

All patients underwent the surgical resection of the tumor followed by histopathological and immunohistochemical examination and tumor staging in relation to TNM classification [12]. Pathologists sub-classified the patients into two groups, low-risk (LR) and high-risk (HR), according to the TNM combined with AFIP classification [9,12]. 

This study was a part of the larger clinical research to evaluate the predictive value of novel advanced imaging modalities in determining the malignant potential of GISTs. Our research was permitted by an Ethical Committee of School of Medicine, University of Belgrade, and written informed consent was obtained from all participants. 

### 2.1. CT and Perfusion CT Technique

CT was performed with the 64-detector row CT (Aquuilion One, Toshiba Medical Systems, Ottawara, Japan). At first, a low-dose unenhanced abdominal CT scan was performed in order to determine the localization of the tumor and plan the CT perfusion study (axial mode, reconstructed slice thickness 8 mm, original slice thickness 0.5 mm, rotation time = 1 s, detector coverage along z-axis 32 mm, reconstructed slices per one rotation 4 × 8 mm, 120 kV, 50 mAs, field of view [FOV] 25 cm, Matrix 512 × 512, and total scan duration 2–3 s). The next series was a low-dose CT perfusion study for which 50 mL of non-ionic iodinated contrast (iopromide, 370 mg/mL iodine) was injected through a 16-G canula in an antecubital vein using a flow rate of 5 mL/s with a chaser bolus of 50 mL saline solution. The perfusion sequence started after a delay of 5 s with respect to the start of intravenous contrast material injection. Four consecutive 8 mm-thick reconstructed sections (a total of 32 mm z-axis coverage), which were previously chosen in the unenhanced series, were scanned every 2 s by using the cine mode acquisition with total scan duration of 50 s (a total of 104 slices per CT study). Additional imaging parameters were tube voltage 100 kV; tube current-time product 50 mAs; and FOV 25 cm; matrix 512 × 512. Patients were instructed to breathe quietly during the scanning. After a pause of fifteen minutes, the third series was performed, which included the conventional CT of the thorax and the abdomen after the intravenous administration of 60–100 mL iodinated contrast agent in the portal venous phase (helical-mode 0.5 mm section thickness, 120 Kv, 120–750 mAs in the tube current modulation mode, 39.5 mm/s table speed, 0.7 s rotation time, 50-cm scan FOV, scan delay 60–70 s and 1-mm reconstructed sections).

### 2.2. CT and Perfusion CT Image Analysis

CT imaging features were analyzed as follows: maximal diameter in the axial slice and tumor structure, which was classified into cystic and solid/necrotic. The shape was depicted as regular or irregular. Tumor localization considered the body of the stomach, antrum or pyloric region. Mucosa was observed in two ways: intact/continuous or disrupted. Growth patterns were categorized as exophytic/mixed or endophytic. The degrees of enhancement of the solid part of tumor tissue were divided into weak, moderate or obvious enhancements. The presence of enlarged feeding or draining vessels (EFDV) was also recorded, as was the presence of metastasis in liver and other organs ( Figure 1, Figure 2 and Figure 3).

Perfusion CT data were analyzed by calculating perfusion according to the deconvolution method. The arterial input function was obtained from a 4–6 mm² circular region of interest (ROI) that was placed in the abdominal aorta. The arterial time-attenuation curve was derived automatically, and parametric colored maps were displayed for each of the four consecutive series of perfusion CT (Figure 4a). One radiologist placed a circular ROI as large as possible within the solid tumor region as well as in the nearby paravertebral skeletal muscles, taking care to avoid large vessels, at the reference and parametric images on each of the four consecutive slices (Figure 4b).

Color parametric maps of the following quantitative perfusion parameters have been automatically computed within these ROIs using the commercial software (Body Perfusion 4.0, GE Health-Care Technologies, USA): blood flow—BF (mL/min/100 g tissue); blood volume—BV (mL/100 g tissue); mean transit time—MTT (s); permeability surface area product—PS (mL/min/100 g tissue); and time to peak—TTP (s) (Figure 5). The values of the perfusion parameters of tumor were recorded for each section of gastric tumor ROI, and the same was done for muscle ROIs. Mean values of the perfusion parameters derived from the four consecutive sections were averaged and used for further analysis.

The effective radiation dose for CT perfusion study was 3.78 mSv (DLP 252 mGy cm), the dose of individualized abdominal CT protocol (unenhanced plus contrast-enhanced portal-venous phase) was 6.32–7.11 mSv (DLP 421.3–474 mGy cm), and the total radiation dose was 10.10–10.89 mSv. The radiation dose for every CT series was calculated from the dose-length product (DLP) listed on the patient dose report, multiplied by the abdominal coefficient factor of 0.015 (mSv mGy^−1^ cm^−1^).

### 2.3. MRI Technique

Patients underwent MRI at 1.5-T (Signa HDxt, GE Healthcare, Waukesha, WI, USA). All images were obtained using an eight-channel phased-array abdominal coil and spine array coil to optimize signal-to-noise ratio (SNR). A routine breath-hold T2-weighted (T2W) single-shot fast spin echo sequence and a breath-hold T2W fat-suppressed (FS) sequence was performed. All DWI examinations were obtained with respiratory-triggered single-shot spin echo-planar imaging with multiple b values (0, 10, 25, 50, 100, 200, 500 and 800 s/mm²). 

### 2.4. Processing of DWI Images and Calculation of ADC and IVIM Parameters

The diffusion weighted images were analyzed by a radiologist with 6 years of experience. T2-weighted images were used to help to determining the localization of the tumor more clearly and avoid necrotic areas (Figure 6). On diffusion-weighted images for b = 0, ROI were positioned within the solid tumor region, on three contiguous axial sections at the place of the largest tumor diameter and then automatically copied to all higher b-values parametric maps. Quantitative ADC maps were calculated on voxel-by-voxel basis on the commercial workstation by using the mono-exponential model for the combination of b = 0 and b = 800 [15]. 

The data for IVIM parameters: true diffusion coefficient (D slow), pseudo-diffusion coefficient (D fast), and perfusion fraction (f) were calculated based on the bi-exponential model by using the MITK Diffusion software, after fitting the signal values from manually located ROIs [15]. 

The final values of D slow, D fast, f, and the ADC for b = 0 and 800 s/mm² were calculated by averaging the three measurements (Figure 6) [15].

### 2.5. Pathological Analysis and Risk Stratification of Gastric GIST’s

The standard treatment of localized primary GISTs without metastasis is surgical resection. Surgical procedures include “wedge resection”, gastric resection and total gastrectomy. After surgical resection, tumor specimens were fixed in 10% formaldehyde, embedded in paraffin and stained with hematoxylin and eosin for pathological evaluation. GISTs were defined as having either spindle-shaped cells, epithelioid cells or a mixture of these two, with positive immunostaining for c-kit and/or DOG-1. Findings such as location, size, mitotic rate and tumor grade based on TNM classification were also revealed.

The American Joint Committee on Cancer tumor/node/metastasis (TNM) classification for gastrointestinal stromal tumors (GISTs) is in wide use [12,16]. Anatomic stage/prognostic groups correspond to the classification of Miettinen et al (AFIP classification) in which size, mitotic index (MI) and anatomic site are the most important prognostic factors [8]. Tumor grade depends on mitotic index, whereas the cut-off value of 5 or less mitosis visible in 5 square mm or per the 50 HPF (high power fields) was established in discriminating low-grade GIST from high-grade GIST (MI > 5) [8,9,10,11,12]. Furthermore, GIST can be categorized into four different stages based upon the mitotic rate, size, spread to lymph nodes, and distant metastasis [12]. Stage I tumors have a low mitotic rate and no spread to lymph nodes. Stage IA tumors are between 2 and 5 cm, whereas stage IB tumors are between 5 and 10 cm. Stage II tumors have a high mitotic rate but are 5 cm or less and without lymphatic spread. Stage III tumors have a high mitotic rate, no lymphatic spread but are greater than 5 cm in diameter. IIIA tumors are between 5 and 10 cm, whereas IIIB tumors are greater than 10 cm. Stage IV tumor can be of any size or mitotic rate but have spread to lymph nodes or distant metastases, such as in the liver. The risk of progression in GIST is determined based on stage, tumor size and mitotic index according to the TNM combined with AFIP classification [9,12]. Accordingly, in our study stages IA and IB are considered low-risk, stage II is intermediate-risk and stages IIIA and IIIB is high-risk gastric GIST [9,12]. 

We sub-classified the patients with GISTs into two groups: high risk (HR), which included high and intermediate risk tumors, and low risk (LR) [12].

### 2.6. Statistical Analysis

For the assessment of the normal distribution of statistical data, the Shapiro–Wilk test was used. Mean ± standard deviation (SD) or median value was presented for quantitative parameters depending on distribution, so as range from minimum to maximum. 

Difference between the attributive characteristics those were analyzed in conventional CT examinations in HR in comparison with LR GISTs was assessed by Chi-square test or Fisher’s exact test. Difference between the quantitative parameter values in HR in comparison with LR GISTs was assessed by t-test for independent samples or Mann–Whitney’s test, depending on the normality of distribution.

Univariate and multivariate binary logistic regression analysis was used for assessing the attributive characteristics and quantitative parameters, which were significantly predictive for HR GIST and building the preoperative imaging-based prognostic model suggesting the high risk GIST, which further was assessed by ROC analysis. 

Statistical significance level was set at 0.05. Statistical analyses were performed by using the SPSS software (Version 17.0 for Windows; SPSS, Chicago, IL, USA).

## 3. Results

This study included 62 patients with GIST of the stomach (32 male and 30 female, mean age of 63 ± 11, from 27 to 83 years). There were 32 patients with high risk GIST (HR group) and 30 patients with low risk GIST (LR group). There was no significant difference in HR and LR group regarding age (61 ± 11, from 38 to 76 years vs. 65 ± 11, from 27 to 83 years, *p* = 0.163) or gender (15 vs. 17 male and 17 vs. 13 female, *p* = 0.441). 

MI was significantly higher (*p* < 0.001) in HR (median 7, from 1 to 69) than in LR group (median 2, from 0 to 5).

The comparison of the classical CT characteristics of tumors in the HR and LR group are presented in the Table 1.

In all 62 patients CT perfusion was carried out. The comparison of CT perfusion parameter values between the HR and LR group are presented in the Table 2. Comparing high grade and low grade tumors, a significant difference was found for PS (*p* = 0.045) and TTP (*p* = 0.047) (Table 2). 

A subgroup of 45 patients (24 HR and 21 LR) underwent MR-DWI examination in addition to CT with CT perfusion. ADC was obtained using the mono-exponential model for the combination of b = 0 and b = 800. ADC values showed high statistically significant differences among the HR and LR GIST groups (*p* < 0.001) (Table 3)

In 34 patients (14 LR and 20 HR), multiple-b values MR-DWI exam was completed by calculating the IVIM parameter values (D slow, D fast and f) based on the bi-exponential model. There was no statistically significant difference between the HR and LR group in mean IVIM parameter values (Table 3). 

Univariate analysis revealed that greater tumor diameter (*p* < 0.001), cystic structure (*p* < 0.001), irregular margins (*p* = 0.007), irregular shape (*p* < 0.001), disrupted mucosa (*p* < 0.001) and visible EFDV (*p* < 0.001), as well as shorter TTP (*p* = 0.006) and less ADC value (*p* = 0.001), were significant predictors of HR GIST. 

Multivariate analysis extracted the irregular shape (*p* = 0.006) and visible EFDV (*p* = 0.017) as the independent predictive CT features of HR GIST (Table 4). The ROC analysis revealed that multivariate linear regression model, which included the shape of the tumor together with the EFVD, achieved an AUC of 0.869 (0.770–0.967), with a sensitivity of 80.0%, specificity of 93.8% and accuracy of 87.1% in predicting the HR GIST (Figure 7).

## 4. Discussion

Our study demonstrates the importance of the morphological CT features of GISTs, which proved their significance in the risk stratification of these tumors. To our knowledge, there are several studies that investigated CT features in correlation to risk stratification of gastric GIST, however none that examined CT perfusion, MRI diffusion and IVIM as functional imaging methods in this regard [3,17,18,19,20,21]. 

In our study, the age ranged from 27 to 83 years, and the median age was 61 years in HR group, which is not much different from other studies [18]. Although the patients with low-risk tumors were generally older in comparison to patients with high-risk tumors, no significant difference was found. Expectedly, as in other studies, the mitotic index values were higher in the group of HR GISTs [17,18,19].

According to our results, the diameter and cystic structure of the tumor, ill-defined margins, irregular shape, exophytic and mixed growth pattern, disrupted mucosal appearance and presence of enlarged feeding or draining vessels were found to be statistically significant factors for risk stratification in the univariate analysis. The logistic regression showed that only irregular tumor shape and presence of EFDV were independent predictors for high metastatic potential. 

Our study applied TNM classification and AFIP criteria that are most commonly used. All classification systems include the size of the tumor lesion as an important predictive factor in the assessment of risk of malignancy [7,8,9,10]. Many studies have shown the statistical significance of diameter in correlation with the tumor risk grade [17,18,19,22,23]. The cut-off for the size of 5 cm was established with GISTs with diameter less than 5 cm having the best prognosis [7,8,9,10,11,12]. We also found that tumor size is a very important predictor of high-risk tumors with an approximate cut-off value of 7 cm between the LR and HR group in our series. Zhou et al. showed similar results to ours that independent predictive factors for the risk stratification of GISTs were tumor size, EFDV and mixed growth pattern [17]. Tateishi et al. reported that tumor diameter larger than 11 cm, the presence of metastases and the invasion of the surrounding structures were the most significant predictors of the high grade malignant potential of these tumors [18]. Kim et al. proved that tumor size was the only CT finding as a significant predictor of tumor aggressiveness [19]. Similarly to our results, Li et al. showed a significant correlation of tumor margins, growth pattern, mucosal appearance and structure with the metastatic risk of GISTs using modified NIH criteria [20]. Contrary to our results, the pattern of enhancement in their study also showed a significant association with pathological risk categories [20]. Tumor size and attenuation are also very important CT features in evaluation of response to chemotherapy [21]. According to the Choi criteria, partial response to imatinib therapy is demonstrated by 10% decrease of tumor size and 15% decrease in tumor attenuation at follow-up CT [21]. In addition to the morphological CT features of the tumor that we analyzed, Grazzini et al. showed that the intratumoral foci of hemorrhage are also statistically associated with the high AFIP risk categories [22]. Chen et al. found that tumor size bigger than 5 cm, exophytic or mixed growth pattern on CT, and tumor cystic degeneration were independent indicators of higher grade tumors [23,24].

It is known that CT perfusion parameters may correlate with tumor risk. It was shown that poorly differentiated tumors have higher values of BF, BV and PS and shorter MTT than less aggressive tumors of the same type [13,25,26,27]. There are no available studies that have investigated the significance of CT perfusion in the preoperative prediction of the metastatic potential of gastric GISTs. Our results also show a tendency for higher values of perfusion parameters in highly aggressive gastric GISTs, as well as shorter MTT. However, there was no statistically significant difference except for PS and TTP. The velocity of inflow to the tumor microvasculature and capillary permeability proved to be the most reliable perfusion parameters for predicting the HR tumor with statistically significantly higher values in the group of HR GISTs, which was also confirmed by other perfusion studies on the different types of malignant tumors [26]. The TTP, indicating the time interval from the beginning of contrast administration to the peak attenuation of the tumor tissue, although a semi quantitative perfusion parameter showed a statistical significance in discriminating the HR from LR GISTs, which was proved by the univariate regression analysis. This result seems significant because there is no need for the commercial CT perfusion software to assess the TTP. The values of this perfusion parameter are available by simply reading from the time-attenuation curve of applied ROI in the CT perfusion series (Figure 4a, b), without complex calculations those are necessary for obtaining the other CT perfusion parameter values. 

Of all investigated quantitative functional parameters in our study, we found that ADC showed the most significant value in predicting the HR of gastric GISTs. Highly significant difference in mean ADC was proved, with significantly lower ADC values in the HR GIST group (Table 3). Yoo et al. investigated the significance of MRI and 18FDG–PET/CT in the differentiation of gastric GIST from non-GISTs [23]. Our results are in line with those of this study, which reported that low ADC was a significant finding as it suggests HR GISTs [23]. Yu et al. also reported that low mean ADC values might be significant in predicting the high malignant potential of GISTs [28]. On univariate logistic regression analysis that was performed in our study, the ADC showed significant potential in differentiating the HR from LR gastric GISTs. 

In oncology, the multiple-b-values diffusion-driven IVIM method in MRI has already shown its potential for the differential diagnosis of malignant and benign tumors, as well as promising prognostic and treatment monitoring biomarker [15,29]. IVIM-derived parameters, in addition to ADC, have shown significance in the differentiation of the most common hypo vascular malignant lesions of the liver [14]. As far as we know, there were no studies investigating the role of IVIM in the differentiation of HR from LR GISTs. We did not find statistically significant differences in IVIM parameters in this regard, which may be a consequence of the relatively small patient sample. Indeed, the true diffusion coefficient (D slow) certainly showed a tendency towards lower values in HR GISTs in comparison to LR (Table 3). To the best of our knowledge, there is only one published study where the therapeutic response of GISTs to imatinib was monitored in mice by using the IVIM methodology [15]. Our results showed a similar relationship between the ADC and IVIM parameters in comparison with their pre-therapeutic findings [15]. 

Our study has several limitations. First, the relatively small patient sample especially for IVIM assessment. Second, our workstations did not have incorporated commercial software for the IVIM analysis, so publicly available external software was used for this purpose. Third, MR-DWI analyses were missing in some patients. Last, our study did not include a follow-up of the involved patients.

In summary, our research resulted in a regression model where the irregular tumor shape and presence of EFDV were the most significant and independent predictors for the high metastatic potential of gastric GISTs. This result shows that morphological characteristics of the tumor detected by conventional CT examination still hold the greatest value in the preoperative risk stratification of gastric GIST. A significant statistical difference was shown regarding the functional CT perfusion parameters TTP and PS and especially the MR-DWI parameter ADC, which, together with the already mentioned classical CT features, can contribute to the more reliable prognosis of the biological behavior of GIST. Therefore, we may conclude that incorporating the advanced functional imaging techniques in the imaging protocol, together with the good knowledge and careful analysis of spectrum of classical morphological CT features, may enable the accurate preoperative assessment of risk stratification of gastric GIST, which is of great importance for the appropriate treatment planning.

## Figures and Tables

**Figure 1 diagnostics-12-02841-f001:**
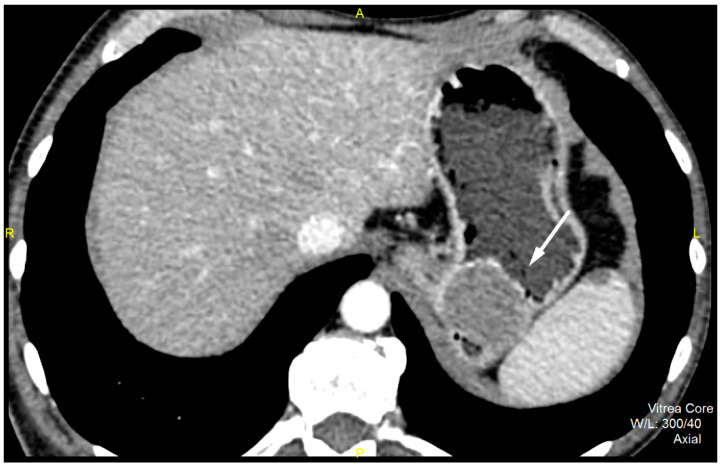
Contrast enhanced computed tomography of the abdomen, axial view, demonstrates low grade GIST (white arrow) in a 54-year old male patient as submucosal, with a round lesion with endophytic growth in the subcardial region of gastric body. The tumor has an approximate diameter of 45 mm, a solid structure with low post-contrast opacification, covered by an intact mucosa.

**Figure 2 diagnostics-12-02841-f002:**
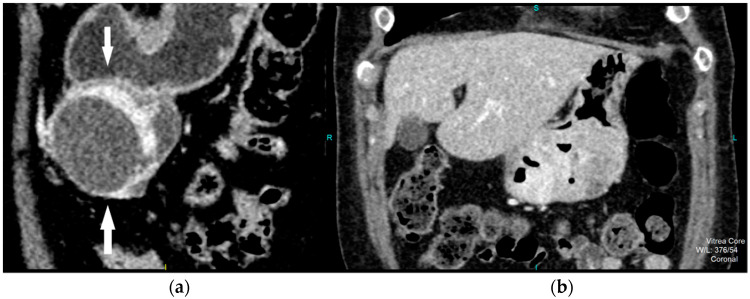
Contrast enhanced CT, coronal view, shows high risk GIST in a 69-year-old female patient, with a clearly demarcated submucosal lesion in the antrum of the stomach with an exophytic growth pattern and predominantly cystic structure with the strong post-contrast enhancement of solid part of tumor (arrows) (**a**). CT examination, coronal section, in a 49-year old female patient reveals intraluminal predominantly solid tumor lesion with irregular shape and margins, endophytic growth with the discontinuity of gastric mucosa and obvious post-contrast enhancement in a HR GIST (**b**).

**Figure 3 diagnostics-12-02841-f003:**
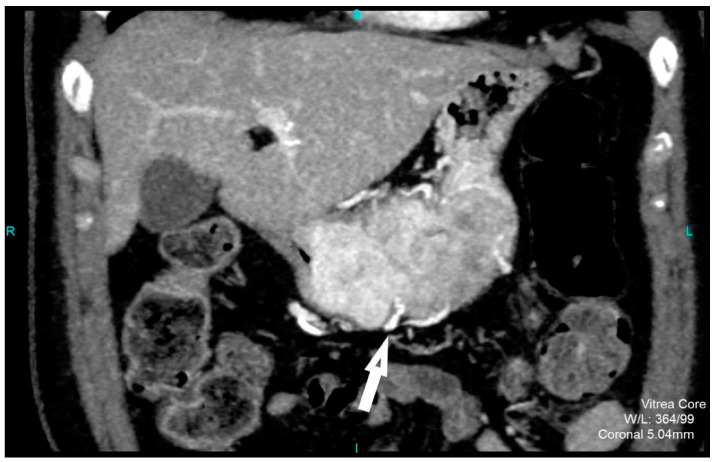
Contrast enhanced CT exam, coronal view, in a 49-year old female patient shows irregular shape and EFDV (white arrow) of a HR GIST.

**Figure 4 diagnostics-12-02841-f004:**
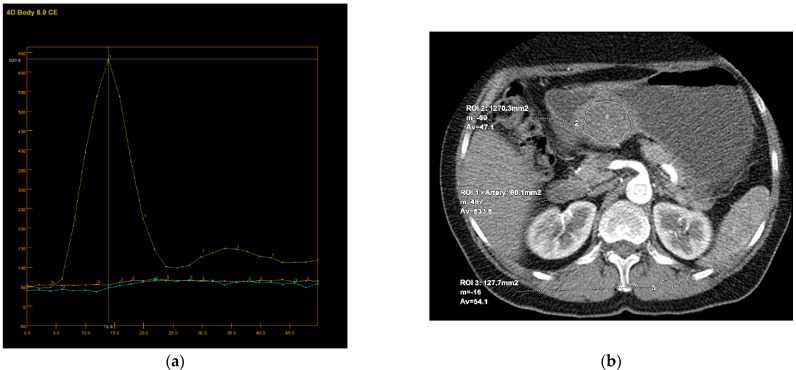
An 8-mm reconstructed image from the low-dose CT perfusion series of a 57-years old female patient with gastric GIST. Time-density curve on computed tomography perfusion (**a**). The arterial (yellow), tumor (blue), and muscle (orange), time-attenuation curves of the corresponding CT perfusion section. A freehand ROI (this tool measures area in square milimeters) in the region of tumor and round ROIs in the aorta and paravertebral muscle (**b**).

**Figure 5 diagnostics-12-02841-f005:**
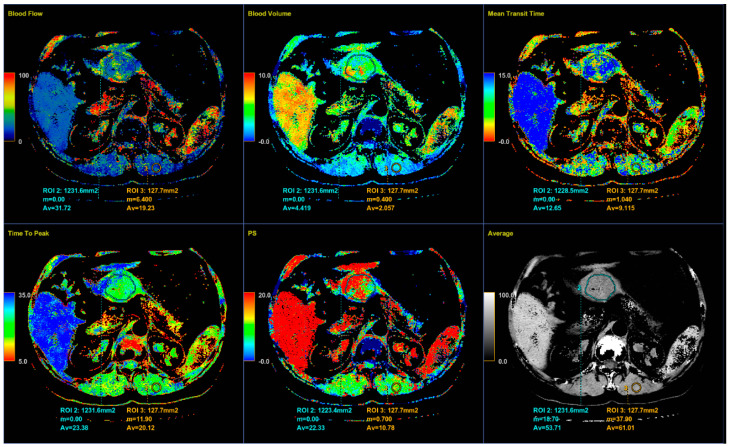
Color parametric maps of the same tumor and muscle ROI (area in square millimeters) automatically computed by the commercial deconvolution-based CT perfusion software. Axial CT sections show the average values of perfusion parameters BF, BV, MTT, TTP, PS, and average density measured for tumor and muscle.

**Figure 6 diagnostics-12-02841-f006:**
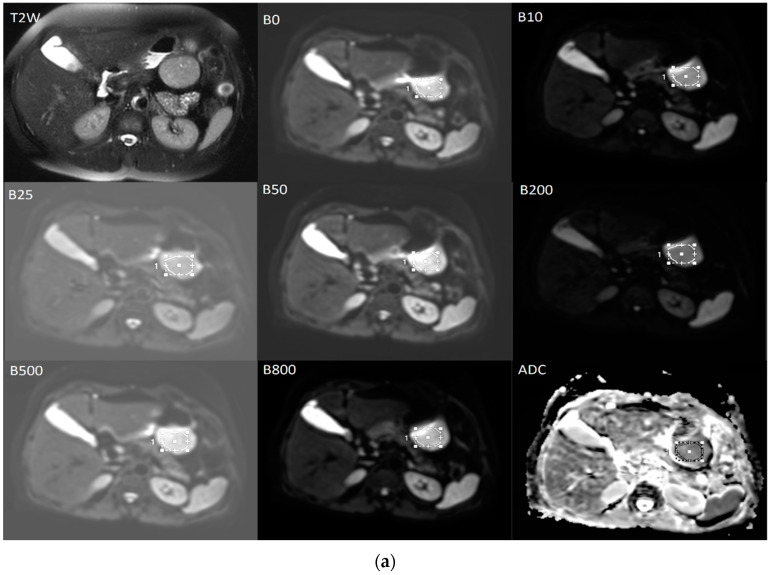
The averaged signal intensity decay and representative IVIM DW images. T2-weighted image in axial view is used for the precise localization of the tumor. DWI with multiple b-values and ROIs (**a**) followed by signal attenuation curve (**b**) where true diffusion coefficient (D slow), pseudo-diffusion coefficient (D fast) and perfusion fraction (f) were calculated based on the bi-exponential model.

**Figure 7 diagnostics-12-02841-f007:**
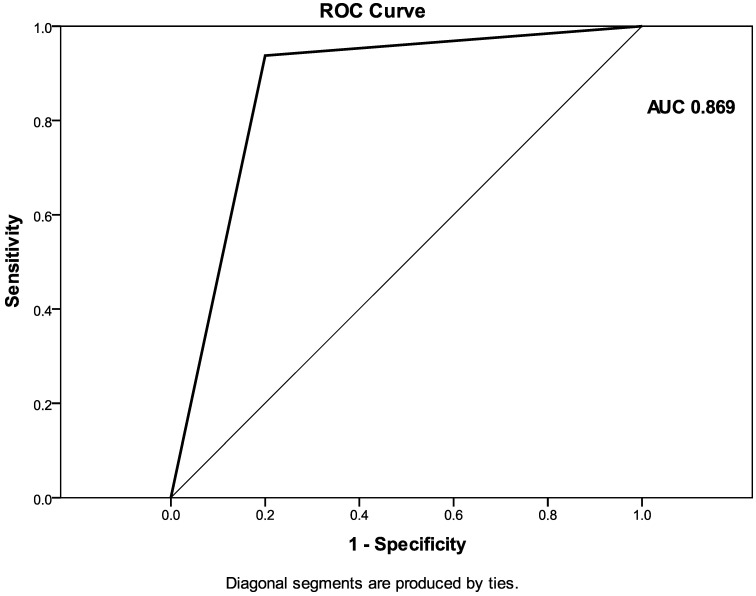
The ROC curve of multivariate regression model with two independent predictors of HR GIST (Probability (0–1; cut-off 0.5) = 0.437 × shape (2-irregular/1-regular) + 0.499 × EFDV (1-present/0-absent)–0.331). AUC = 0.869 (0.770–0.967).

**Table 1 diagnostics-12-02841-t001:** Classical CT characteristics of gastric GIST in the HR and LR group.

CT Characteristics of Gastric GIST		LR Group (n = 30)	HR Group (n = 32)	*p* Value
Diameter (mm)		60(15–94)	115(40–340)	<0.001
Localization	Body	12	16	0.034
	Antrum	16	8	
	Pylorus	2	8	
Margins	1-well-defined	29	24	0.016
	2-ill-defined	1	8	
Growth pattern	1-exophytic/mixed	19	29	0.015
	2-endophytic	11	3	
Tumor enhancement	Low	5	9	0.477
	Moderate	19	19	
	High	6	4	
Shape	1-regular (round)	28	10	<0.001
	2-irregular	2	22	
Structure	1-solid/necrotic	23	8	<0.001
	2-cystic	7	24	
Mucosa	1-continuous	25	12	<0.001
	2-discontinuous	5	20	
	(ruptured)			
Enlarged feeding or draining vessels (EFDV)	0-absent	26	6	<0.001
	1-present	4	26	

**Table 2 diagnostics-12-02841-t002:** CT perfusion parameter values of gastric GIST in the HR and LR groups.

Perfusion Parameter	LR GIST (n = 30)	HR GIST (n = 32)	*p*
BF (mL/min/100 g)	51.1 (12.5–177.5)	57.5 (27.3–148.5)	0.522
BV (mL/100 g)	4.8 ± 1.7 (1.9–9.2)	4.9±1.5	0.746
MTT (s)	8.4 ± 3.2 (3.7–11.6)	7.5 ± 2.5 (3.2–16.6)	0.201
PS (mL/min/100 g)	24.9 (10.7–85.5)	34.2 (15.1–91.6)	0.045
TTP (s)	28.7 ± 5.2 (19.0–39.0)	25.8 ± 6.0 (17.0–40.0)	0.047

**Table 3 diagnostics-12-02841-t003:** MR-DWI (ADC and IVIM) parameter values of gastric GIST in HR and LR group.

MR-DWI Parameter (n = 45)	LR GIST (n = 21)	HR GIST (n = 24)	*p* Value
ADC (mm²/s) (n = 21)	0.00195 (0.001–0.003)	0.00150 (0.00098–0.00302)	<0.001
**IVIM paremeters (n = 34)**	**LR group (n = 14)**	**HR group (n = 20)**	***p* value**
f (%)	26.6 (7.1–86.0)	29.4 (13.7–73.0)	0.545
D fast (mm²/s)	0.0192 (0.0054–0.0686)	0.0153 (0.0050–0.0709)	0.323
D slow (mm²/s)	0.00170 ± 0.00065 (0.00036–0.00270)	0.00134 ± 0.00053 (0.00031–0.00279)	0.081

**Table 4 diagnostics-12-02841-t004:** Significant predicting parameters of HR GIST according to the multivariate logistic regression analysis.

Model		Unstandardized Coefficients		Standardized Coefficients	t	Sig.
		B	Std. Error	Beta		
1	(Constant)	−0.331	0.123		−2.692	0.009
	Shape 2-irregular 1-regular (oval)	0.437	0.092	0.426	4.746	0.000
	EFDV 1-present 0-absent	0.499	0.090	0.499	5.561	0.000

## Data Availability

Not applicable.

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
