# Peer review of "The Utility of Conventional CT, CT Perfusion and Quantitative Diffusion-Weighted Imaging in Predicting the Risk Level of Gastrointestinal Stromal Tumors of the Stomach: A Prospective Comparison of Classical CT Features, CT Perfusion Values, Apparent Diffusion Coefficient and Intravoxel Incoherent Motion-Derived Parameters"

_diagnostics, 2022, doi:10.3390/diagnostics12112841_

Round 1
Reviewer 1 Report
This study evaluates the role of classical CT, CT-perfusion, and MR-DWI in predicting the histological grade of gastric GIST. Preoperative identification of patients with high-risk features would be of interest for clinical decision-making in a multidisciplinary approach, particularly for neoadjuvant therapy. The study is well conducted, the methods are adequately used, and the results mainly sustain the conclusions. Thus, the results of the present study would potentially add value to the current literature. Furthermore, the paper is well-written and well-organized. In conclusion, the paper would be of interest to journal readers. Only a few modifications should be made before potential acceptance for publication.
Define abbreviations at their first use in the text.
Figures 7-9 appear to be redundant.
Reviewer 2 Report
The author investigated the potential role of standard and advanced CT and MR imaging techniques as classical CT features, CT-perfusion, DWI-ADC and IVIM related parameter values in non-invasive prediction of histopathologic risk grade of gastric GISTs. However, some major issues need to be address before this article to be published.
1. Does the number of patients enough to investigate the utility of conventional CT, CT perfusion and quantitative diffusion-weighted imaging?
2. The patients are from 27 to 83 years in this paper, does the age impact the results?
Reviewer 3 Report
Manuscript entitled "The utility of conventional CT, CT perfusion and quantitative diffusion-weighted imaging in predicting the high grade of gastrointestinal stromal tumor of the stomach"
Major defects:
1. The classification of HR and LG is not convincing. The authors should WHO/AJCC classification to define its risk. At the same time, the authors must clearly describe the mitosis of the tumor.
2. The correlation study is too rough. More clinicopathologic variables should be added to make this work readable.
